# 7-Epitaxol Induces Apoptosis and Autophagy in Head and Neck Squamous Cell Carcinoma through Inhibition of the ERK Pathway

**DOI:** 10.3390/cells10102633

**Published:** 2021-10-02

**Authors:** V. Bharath Kumar, Ming-Ju Hsieh, B. Mahalakshmi, Yi-Ching Chuang, Chia-Chieh Lin, Yu-Sheng Lo, Hsin-Yu Ho, Jen-Tsun Lin

**Affiliations:** 1Department of Medical Laboratory Science and Biotechnology, Asia University, Taichung 413, Taiwan; bharathvel@gmail.com; 2Oral Cancer Research Center, Changhua Christian Hospital, Changhua 500, Taiwan; 170780@cch.org.tw (M.-J.H.); 177267@cch.org.tw (Y.-C.C.); 181327@cch.org.tw (C.-C.L.); 165304@cch.org.tw (Y.-S.L.); 3College of Medicine, National Chung Hsing University, Taichung 402, Taiwan; 4Graduate Institute of Biomedical Sciences, China Medical University, Taichung 404, Taiwan; 5Department of Research and Development, Vels Publishers, Bodinayakanur 625513, Tamilnadu, India; mahalakshmibharath05@gmail.com; 6Division of Hematology and Oncology, Department of Medicine, Changhua Christian Hospital, Changhua 500, Taiwan; 7School of Medicine, Chung Shan Medical University, Taichung 402, Taiwan

**Keywords:** 7-Epitaxol, head and neck cells, apoptosis, autophagy

## Abstract

As the main derivative of paclitaxel, 7-Epitaxol is known to a have higher stability and cytotoxicity. However, the anticancer effect of 7-Epitaxol is still unclear. The purpose of this study was to explore the anticancer effects of 7-Epitaxol in squamous cell carcinoma of the head and neck (HNSCC). Our study findings revealed that 7-Epitaxol potently suppressed cell viability in SCC-9 and SCC-47 cells by inducing cell cycle arrest. Flow cytometry and DAPI staining demonstrated that 7-Epitaxol treatment induced cell death, mitochondrial membrane potential and chromatin condensation in OSCC cell lines. The compound regulated the proteins of extrinsic and intrinsic pathways at the highest concentration, and also increased the activation of caspases 3, 8, 9, and PARP in OSCC cell lines. Interestingly, a 7-Epitaxol-mediated induction of LC3-I/II expression and suppression of p62 expression were observed in OSCC cells lines. Furthermore, the MAPK inhibitors indicated that 7-Epitaxol induces apoptosis and autophagy marker proteins (cleaved-PARP and LC3-I/II) by reducing the phosphorylation of ERK1/2. In conclusion, these findings indicate the involvement of 7-Epitaxol in inducing apoptosis and autophagy through ERK1/2 signaling pathway, which identify 7-Epitaxol as a potent cytotoxic agent in HNSCC.

## 1. Introduction

Head and neck cancer is the sixth most common type of cancer worldwide [1]. The majority of head and neck cancers that originate from the mucosal epithelial lining of the oral cavity, larynx, and pharynx are collectively called head and neck squamous cell carcinoma (HNSCC) [2]. In 2018, about 890,000 new cases of HNSCC, including 450,000 deaths, were registered globally [3]. Among various types of HNSCCs, carcinomas in the oral cavity and larynx are primarily caused by tobacco smoking/chewing and excessive alcohol consumption. By contrast, pharynx cancers are mainly associated with human papillomavirus (HPV) infection [4,5]. While HPV-negative HNSCC is mostly prevalent in southeast Asia, HPV-positive carcinoma cases are mostly found in the United States and western Europe [4,5].

Regarding the pathogenesis of HNSCC, there is evidence indicating that metabolic activation of tobacco-derived human carcinogens causes the generation of reactive free radicals, which subsequently damage DNA and disrupt genetic integrity. A gradual accumulation of mutations in tumor suppressor genes (p53 and PTEN) and/or genes of signaling pathway components (AKT–mTOR pathway and RAS–MAPK pathway) is the primary cause of HPV-negative HNSCC [6]. In addition, increased expressions of receptor tyrosine kinases (EGFR) or aberrant signaling of major regulators of oxidative stress (NRF2 and KEAP1) can lead to the development of HNSCC [7,8]. 

The gold standard treatment strategies for HNSCC include surgery, radiotherapy, and chemotherapy. In patients with non-metastatic small primary tumors, a cure rate of more than 80% can be achieved by resection or radiation. Similarly, in high-risk patients with recurrent or metastatic cancer, postoperative concurrent application of radiotherapy and chemotherapy has been shown to increase disease-free survival [9]. With the advancement in treatment strategies, the survival rate of HNSCC has increased from 55% in 1996 to 66% in 2006. However, people suffering with larynx cancer and elderly people in general are still associated with a poor HNSCC-related survival rate [10]. 

Among various chemotherapeutic agents, paclitaxel is widely used as a cytotoxic and apoptosis-inducing agent in many cancer types, including ovarian, breast, brain, and lung cancers [11,12,13,14]. Paclitaxel, also known as Taxol, is a naturally occurring tricyclic diterpenoid compound found in the evergreen tree, *Taxus brevifolia* [15]. Regarding its mode of action, paclitaxel is known to inhibit cell cycle progression, mitosis, and cancer cell proliferation by stabilizing microtubules [16]. However, there is an alternative hypothesis suggesting that paclitaxel causes cancer cell death by inducing multipolar divisions [17].

The major active metabolite of paclitaxel is 7-Epitaxol, which is more cytotoxic to cancer cells and more thermodynamically stable [18,19]. It is an epimerization product of taxol, although it shows no difference in the structure of chemical formulas with taxol, but the functional groups [OH^−^] and [H^+^] substitute the side chain for biological activity, thus affecting change [19]. Baccatin III produced is the main cause of minor compounds and has biological activity, from taxol sensitivity to alkaline hydrolysis [20]. Paclitaxel with multiple hydrolytically sensitive ester groups has a chiral center that rapidly undergoes epimerization, leading to the production of 7-Epitaxol, which works similarly to paclitaxel in both in vitro and in vivo conditions [18,19,20]. 

Despite having higher anticancer potency and biological stability than paclitaxel, the impact of 7-Epitaxol as a potent chemotherapeutic agent has not been studied widely. The present study was designed to evaluate the anticancer effects of 7-Epitaxol on HNSCC cell viability, as well as to determine the mode of action of 7-Epitaxol.

## 2. Materials and Methods

### 2.1. Chemical

We purchased 7-Epitaxol (7-E) (≥98% purity) from ChemFaces (Wuhan, Hubei, China), and it was dissolved in dimethyl sulfoxide (DMSO) to prepare 100 mM stock solution, which was further diluted to prepare working solutions of 0 (vehicle group), 50, 100, and 200 nM concentrations. The final concentration of DMSO in the working solutions was less than 0.2%. Other chemical reagents used in the study, including 3-(4,5-dimethylthiazol-2-yl)-2,5-diphenyltetrazolium bromide (MTT), propidium iodide (PI), RNase A, DAPI dye, protease inhibitor cocktail, and phosphatase inhibitor cocktail, were obtained from Sigma-Aldrich (St Louis, MO, USA). The primary antibodies against cyclin A, cyclin B, CDK2, CDK4, FAS, DR5, DcR3, DcR2, cleaved caspase-3, -8, -9, cleaved poly (ADP-ribose) polymerase (PRAR), Bax, Bak, Bcl-xL, Bcl-2, Bid, LC3-I/II, p62, p-AKT, AKT, p-ERK1/2, ERK1/2, p-p38 MAPK, p38 MAPK, p-JNK1/2, JNK1/2, and β-actin were purchased from Cell Signaling Technology (Danvers, MA, USA). Specific inhibitor for ERK1/2 (U0126) was purchased from Santa Cruz Biotechnology (Santa Cruz, CA, USA).

### 2.2. Cell Culture

Two HNSCC cell lines, SCC-9 (ATCC, Manassas, VA, USA) and SCC-47 (Merck Millipore; Burlington, MA USA), were selected for the experiments. The HNSCC cell lines were cultured in Dulbecco’s Modified Eagle Medium (DMEM; Life Technologies, Grand Island, NY, USA) supplemented with 10% fetal bovine serum, 0.1 mM nonessential amino acids, 1 mM glutamine, 1% penicillin/streptomycin (10,000 U/mL penicillin and 10 mg/mL streptomycin), 1.5 g/L sodium bicarbonate, and 1 mM sodium pyruvate. The cells were maintained at 37°C in a humidified atmosphere of 5% CO_2_.

### 2.3. Cell Cytotoxicity 

The cells were cultured in 96-well plates at a density of 1 × 10^4^ cells/well overnight, followed by incubation with different concentrations of 7-Epitaxol (0, 50, 100, or 200 nM) for 24, 48, or 72 h. Upon completion of the treatment, 20 µL of MTT (5 mg/mL) solution was added to each well and incubated for 4 h at 37°C. The blue formazan crystals formed were dissolved in DMSO and the absorbance was measured at 595 nm using spectrophotometry. The entire procedure was repeated three times using the same conditions to obtain three independent experimental replicates.

### 2.4. Colony Formation Assay

The SCC-9 and SCC-47 cell lines were seeded onto 6-well plates at a density of 5 × 10^3^ cells/well and cultured overnight, followed by incubation with different concentrations of 7-Epitaxol (0, 50, 100, and 200 nM). The incubation medium was changed every 3 days. After two weeks, the colonies were fixed with 4% paraformaldehyde and then stained with 0.3% crystal violet solution. The stained colonies were dissolved in DMSO and counted by a stereomicroscope as previously described [21].

### 2.5. Cell Cycle Analysis

The SCC-9 and SCC-47 cell lines were seeded onto 6-well plates at a density of 5 × 10^5^ cells/well and cultured overnight. The cells were next incubated with different concentrations of 7-Epitaxol for 24 h. Afterwards, the cells were collected, fixed in 70% ice-cold ethanol overnight, and stained with PI buffer (4 mg/mL PI, 1% Triton X-100, 0.5 mg/mL RNase A in PBS) for 30 min in the dark at room temperature. Cell cycle distribution was analyzed by BD Accuri C6 Plus flow cytometry (BD Biosciences, San Jose, CA, USA) and the data were analyzed using BD CSampler Plus software (BD Biosciences, San Jose, CA, USA).

### 2.6. Western Blot Analysis 

The HNSCC cells were first treated with different concentrations of 7-Epitaxol for 24 h, followed by lysis with RIPA buffer containing protease/phosphatase inhibitor cocktails to obtain cellular proteins. After measuring protein concentrations using a BCA (Thermo Fisher Scientific) assay, the samples were separated using SDS-PAGE and transferred to PVDF membranes (Millipore, Bedford, MA, USA). The membranes were then blocked with 5% nonfat milk in TBST for 1 h, followed by incubation with appropriate primary antibodies (dilution ratio 1:1000) overnight at 4 °C. The protein bands were visualized using enhanced chemiluminescence with an HRP substrate (Millipore).

### 2.7. Annexin V/PI Double Staining Assay 

As previously described [22], the SCC-9 and SCC-49 cell lines were treated with different concentrations of 7-Epitaxol for 24 h. Then, the cells were harvested and suspended in PBS (2% BSA) and incubated with Muse Annexin V and Dead Cell reagent (EMD Millipore, Billerica, MA, USA) for 20 min at room temperature in the dark. The data were analyzed by Muse Cell Analyzed flow cytometry (Merck Millipore, Burlington, MA, USA).

### 2.8. DAPI Staining

The cells were cultured in an 8-well glass chamber slide at a density of 1 × 10^4^ cells/well overnight, followed by treatment with different concentrations of 7-Epitaxol for 24 h. Afterward, the cells were collected, fixed by 4% formaldehyde for 30 min, and stained with DAPI dye (50 ug/mL) for 15 min in the dark. The nuclear morphological changes were assessed in at least 500 cells and photographed using an Olympus FluoView FV1200 Confocal Microscope (Olympus Corporation, Shinjuku, Tokyo).

### 2.9. Mitochondrial Membrane Potential Measurement

As previously described [23], SCC-9 and SCC-47 cells were incubated with different concentrations of 7-Epitaxol for 24 h. The cells were collected and stained with Muse MitoPotential working solution at 37 °C for 20 min. After incubating the cells with 5 μL of 7-AAD for 5 min, a Muse Cell Analyzer flow cytometer (EMD Millipore) was used to detect samples. The data were analyzed by a Muse Cell Analyzer (Millipore).

### 2.10. Detection of Autophagy 

The cells were cultured (1 × 10^4^/well) in 96-well plates overnight and incubated with different concentrations of 7-Epitaxol (0, 50, 100, or 200 nM) for 24 h. After removing the medium, 100 μL of Autophagy Green working solution (Cell Meter Autophagy Assay Kit, AAT Bioquest, Inc., Sunnyvale, CA, USA) was added into each well and incubated for 60 min. After washing the cells 3–4 times, fluorescence intensity was measured with a fluorescence microplate reader at Ex/Em = 485/530 nm. Finally, 20 μL of MTT (5 mg/mL) solution was added to each well to assess cell viability. The respective fluorescence levels were normalized by cell cytotoxicity results.

### 2.11. Statistical Analysis

The experimental data are expressed as means ± standard deviation. Each experiment was replicated at least three times. The statistical analyses were conducted by ANOVA, Tukey’s post hoc test, and Student’s *t*-test. In all cases, a *p* value of <0.05 was considered statistically significant. All statistical analyses were performed using Sigma-Stat 2.0 (Jandel Scientific, San Rafael, CA, USA).

## 3. Results

### 3.1. Cytotoxic Effects of 7-Epitaxol on HNCSS Cell Lines

To investigate the anti-proliferative effects of 7-Epitaxol (7-E), two HNSCC cell lines, SCC-9 and SCC-47, were treated with increasing concentrations of 7-E (0, 50, 100, and 200 nM) for 24, 48, and 72 h and subjected to an MTT assay (Figure 1B,C). The working concentrations of 7-E were based on a previous study that treated paclitaxel on squamous carcinoma cells [24]. The findings of the MTT assay revealed that treatment with 7-E significantly reduced cell viability in a time-dependent manner compared to that in untreated control cells (Figure 1B,C). A similar anti-proliferative impact of 7-E was also observed in the colony formation assay, which revealed that all tested concentrations of 7-E were capable of significantly reducing the colony-forming ability of HNSCC cells (Figure 1D–G). Taken together, these observations indicate that 7-E acts as a potent anti-proliferative agent.

### 3.2. Effect of 7-Epitaxol on Cell Cycle Progression and Apoptosis of HNCSS Cells

To investigate the mechanism by which 7-E exerts its cytotoxic effect, the cell cycle distribution of 7-E-treated HNSCC cells was analyzed using flow cytometry. As observed in Figure 2A,B, the treatment with 7-E caused cell cycle arrest and increased the cell cycle rate at the sub-G1 phase in both NHSCC cell lines. However, in the SCC-47 cell line, 7-E treatment caused an induction in cell cycle rate at the S phase. At the G2/M phase, 7-E treatment caused an induction and a reduction in cell cycle rate in SCC-9 and SCC-47 cells, respectively. Overall, these observations indicate that the effect of 7-E on cell cycle may vary with cell types. 

To further evaluate cell cycle inhibitory effects, 7-E-treated cells were analyzed for cell cycle regulatory proteins. As observed in Figure 2C,D, the 7-E treatment significantly downregulated the expressions of key cell cycle regulators, including cyclin A, cyclin B, and cyclin-dependent kinases 2 and 4 (CDK2 and CDK4) in both cell lines. 

To evaluate whether 7-E can modulate cell viability through apoptosis, the changes in cell morphology and nuclear condensation after 24 h of 7-E treatment were analyzed using DAPI staining. As observed in Figure 3C,D, the apoptosis index increased significantly in 7-E-treated cells in a dose-dependent manner. 

To further evaluate apoptotic phenomena after 7-E treatment, HNSCC cells stained with Annexin V-FITC/PI were sorted by flow cytometry. As observed in Figure 3A,B, the percentage of apoptotic cells in the early apoptotic stage (Annexin V^+^/PI^−^) and late apoptotic stage (Annexin V^+^ and PI^+^) increased significantly and dose dependently after 7-E treatment. At the highest concentration, 7-E induced apoptosis in 49.87% of the SCC-9 cells and 26.74% of the SCC-47 cells.

### 3.3. Effect of 7-Epitaxol on Apoptotic Signaling Pathways

Due to the significant involvement of mitochondria in mediating cell death, the effect of 7-E on mitochondrial membrane potential was initially measured. As shown in Figure 4A,B, 7-E treatment (0–200 nM) significantly increased the percentage of depolarized cells to 13.36%, 22.94% and 28.13% in SCC-9 cells and 15.46%, 17% and 34.57% in SCC-47 cells.

Next, the impact of 7-E on both extrinsic and intrinsic apoptotic pathways was evaluated. As observed in Figure 4C,D, 7-E treatment significantly increased the expression of key proteins of the Fas and tumor necrosis factor (TNF) pathway, including Fas, death receptor 5 (DR5), decoy receptor 3 (DcR3), and DcR2, in both cell lines. Regarding the intrinsic apoptotic pathway, 7-E treatment (200 nM) significantly increased the expressions of pro-apoptotic Bcl-2 family proteins, including Bax, Bak, and Bid approximately 6.5, 3.4, and 1.6-fold change in SCC-9 cells compared to that in untreated control cells, and significantly decreased the expression of anti-apoptotic proteins Bcl-2 and Bcl-xL in SCC-9 and SCC-47 cells, respectively (Figure 5C,D). 

Since activation of caspases is the ultimate step in both intrinsic and extrinsic apoptotic pathways, the expression levels of the cleaved forms of caspases 3, 8, and 9, as well as Poly (ADP-ribose) polymerase (PARP), were determined. The results indicated that, in both cell lines, 7-E treatment (200 nM) significantly increased the expressions of cleaved PARP, caspase-3, caspase-8, and caspase-9 reach in 2.9, 1.6, 4.9, 3.1-fold change individually in SCC-9 cells, and 8.3, 2.6, 5.2, 2.4-fold change in SCC-47 cells compared to that in untreated control cells. (Figure 5A,B).

### 3.4. Effect of 7-Epitaxol on Autophagy Signaling Pathway 

Although autophagy is commonly regarded as a cytoprotective mechanism for maintaining cellular homeostasis, there is a growing body of evidence highlighting the potential involvement of autophagic cell death in tumor suppression. To evaluate the anticancer potential of 7-E beyond apoptosis, a Cell Meter^TM^ Autophagy Assay was performed to examine specific autophagosome markers. As shown in Figure 6A, the green fluorescence levels in 7-E-treated (200 nM) cells increased to 247.23% in SCC-9 cells and 147.78% in SCC-47 cells compared to those in untreated control cells. This indicates the induction of autophagy pathway mediators in 7-E-treated HNSCC cells. 

For further evaluation, the expressions of various autophagy-related proteins were assessed using Western blot. Our findings revealed that 7-E treatment increased the expression of LC3-I/II and reduced the expression of p62 (Figure 6B,C). Taken together, these observations confirm that 7-Epizaxol induces autophagy in HNSCC cell lines.

### 3.5. Effect of 7-Epitaxol on AKT and MAPK Pathways

To identify the signaling cascade associated with 7-E-mediated modulation of cellular apoptosis and autophagy, expression levels of the components involved in the AKT and MAPK signaling pathways were analyzed in HNSCC cells. As observed in Figure 7A,B, 7-E (200 nM) treatment significantly reduced the phosphorylation of AKT (1.3 and 1.01-fold decrease) and ERK1/2 (5.5 and 4.8-fold decrease) in both SCC-9 and SCC-47 cells compared to that in untreated control cells, respectively. Moreover, a significantly increased phosphorylation of JNK approximately 1.8-fold change in 7-E (200 nM)-treated SCC-9 cells and significantly increased phosphorylation of p38 approximately 2.8-fold change in 7-E (200 nM)-treated SCC-47 cells compared to that in untreated control cells, respectively. 

Since the most prominent effect of 7-E was observed in ERK1/2 phosphorylation, this crosstalk was further evaluated in the context of cellular apoptosis and autophagy. For this purpose, the cells were treated with U0126, a potent ERK inhibitor, in presence or absence of 7-E (200 nM) for 24 h. As observed in Figure 7C–F, cotreatment with 7-E and U0126 increased the expressions of cleaved PARP (2.8 and 2.1-fold change), cleaved caspase 3 (3.5 and 1.7-fold change), and LC3-I/II (1.7 and 1.9-fold change) in both SCC-9 and SCC-47 cells compared with 7-E treatment alone. Taken together, these findings suggest that 7-E induces apoptosis and autophagy in HNSCC cells by downregulating ERK1/2 phosphorylation.

## 4. Discussion

The present study describes the anticancer efficacy of 7-Epitaxol, the major active metabolite of paclitaxel, on HNSCC. As observed in the study, 7-Epitaxol exerts significant cytotoxic effects on HNSCC cells (Figure 1) by causing cell cycle arrest and inducing apoptosis and autophagy (Figure 2, Figure 3, Figure 4, Figure 5 and Figure 6). Regarding its mode of action, the study findings indicate that 7-Epitaxol exerts anti-proliferative effects by downregulating AKT and ERK1/2 phosphorylation (Figure 7). 

Being the most effective natural chemotherapeutic drug, paclitaxel has been widely and extensively used as a cytotoxic agent in various cancer types [25,26,27,28,29]. In line with the present study findings, previous studies have shown that paclitaxel significantly reduces the viability of cancer cells by inducing cell cycle arrest and activating apoptotic pathways [29,30]. When administered in combination with other compounds, paclitaxel has shown significantly higher efficacy in inhibiting the growth of cancer cells [31,32]. Given the wide range of toxic side effects of solvent-based paclitaxel preparations, several nanoparticle-based formulations of paclitaxel have been developed, with the aim of improving drug efficacy and reducing treatment-induced adverse events [33,34,35]. For instance, liposome-based paclitaxel formulations have been shown to exert lower levels of neurotoxicity in both in vitro and in vivo conditions compared to standard preparations [36]. Similarly, preparation of a hydrophobic prodrug by conjugating paclitaxel with vitamin E, as well as encapsulating the prodrug into nanoparticles, has been shown to significantly increase the anticancer efficacy of paclitaxel [37]. Taken together, these observations highlight the need for continuous upgradation in paclitaxel-based treatment strategies for better cancer management.

As mentioned earlier, because of its high instability in aqueous solution, the hydroxyl group of paclitaxel at the 7 position rapidly undergoes epimerization, giving rise to 7-Epitaxol, which is more thermodynamically stable and more cytotoxic than paclitaxel [38,39]. In this context, a recent study has revealed that, in standard cell culture conditions, bone marrow-derived mesenchymal stem cells are able to incorporate paclitaxel for targeted cellular delivery. At the site of delivery, these modified stem cells deliver biologically active paclitaxel together with its active metabolite 7-Epitaxol [40]. These findings indicate that 7-Epitaxol is the main metabolite of paclitaxel that possesses equivalent pharmacological activity as paclitaxel. As it has comparatively higher stability and cytotoxicity than paclitaxel, 7-Epitaxol was specifically selected in the present study for evaluation. 

Being a microtubule stabilizer, paclitaxel is known to arrest the cell cycle at the G0/G1 and G2/M phases to induce cancer cell death [41]. This is in line with the present study findings, which show that 7-Epitaxol induces cell cycle arrest in both HNSCC cell lines (Figure 2A,B). Regarding cell cycle checkpoint regulators, 7-Epitaxol caused significant reductions in cyclin A, cyclin B, CDK 2, and CDK4 expression compared to untreated cells (Figure 2C,D). Previous studies investigating the process of cell cycle regulation in cancer cells have shown that loss of cyclin B1 function in cells directly results in downregulation of cyclin A and CDK2, leading to cell cycle arrest and induction of apoptosis [42,43]. These findings indicate that 7-Epitaxol effectively inhibits mitosis in cancer cells by downregulating cell cycle checkpoint proteins. In addition, the primary antitumor mechanism of paclitaxel in tumor cells is to cause a mitotic block by stabilizing microtubules and decreasing the dynamic nature of these cytoskeletal structures [44]. As an anti-mitotic agent, paclitaxel would be expected to inhibit cell proliferation at the G2/M phase of the cell cycle; however, the findings of the present study show that 7-Epitaxol induces cell cycle arrest. The possible effect of 7-Epitaxial in stabilizing the microtubules of tumor cells needs to be further confirmed by relevant research experiments.

Based on our findings, 7-Epitaxol induces HNSCC cell apoptosis (Figure 3) by increasing mitochondrial depolarization and increasing the expressions of FAS and death receptors (Figure 4). In addition, increased expressions of pro-apoptotic proteins Bax, Bak, and Bid, decreased expressions of anti-apoptotic proteins Bcl-2 and Bcl-xL, and increased activation of PARP and caspases 3, 8, and 9 were observed in 7-Epitaxol-treated HNSCC cells (Figure 5). These findings are in line with previous studies demonstrating that paclitaxel induces cancer cell apoptosis by increasing pro-apoptotic protein expression, reducing anti-apoptotic protein expression, and subsequently activating PARP and caspase 3 [45,46]. Taken together, these findings indicate that paclitaxel and its metabolite 7-Epitaxol share similar biological activities. 

Interestingly, there is evidence indicating that the experimental upregulation of cellular autophagy increases cancer cell sensitivity to paclitaxel cytotoxicity [47]. In the present study, 7-Epitaxol was found to increase autophagy by increasing the expression of LC3-I/II and reducing the expression of p62 (Figure 6). Mechanistically, the p62 protein directly interacts with LC3-I/II to form autophagosomes, which are required for the degradation and recycling of damaged cellular components via autophagy [48]. An upregulation of the most potent autophagy marker, LC3-I/II, in response to 7-Epitaxol clearly indicates an induction of an autophagic pathway in HNSCC cells (Figure 6). 

The presence of multiple domains places p62 at the center of various cellular processes, including cell survival, apoptosis, and autophagy [49]. In many cancer types, the silencing of p62 has been found to significantly reduce cell proliferation and induce autophagy [50]. In lung cancer cells, ginkgetin, a flavonoid compound, has been found to induce autophagic cell death by suppressing the expression of p62 [51]. These findings are in line with the present study, in which 7-Epitaxol was found to induce autophagy by suppressing p62 expression (Figure 6). 

Aberrant activation of MAPK and AKT/PI3K signaling pathways has been observed in many cancer types [52,53]. Given the significant involvement of these pathways in regulating cell survival, differentiation, and apoptosis, the effect of 7-Epitaxol on the key signaling components, including AKT, ERK1/2, p38, and JNK1/2, was assessed. Our findings revealed that 7-Epitaxol significantly reduced the phosphorylation of ERK1/2 in both HNSCC cell lines (Figure 7A,B). Furthermore, the cotreatment of cells with an ERK inhibitor and 7-Epitaxol caused a further induction of apoptotic and autophagic markers compared to that caused by 7-Epitaxol treatment alone (Figure 7C–F). These findings clearly indicate that 7-Epitaxol induces apoptotic and autophagic cell death in HNSCC by suppressing the ERK1/2 signaling pathway. In support of these findings, some recent studies have shown that, in paclitaxel-resistant cancer cells, combination therapies with paclitaxel and phytochemicals can make cancer cells sensitive to paclitaxel by suppressing the ERK1/2 signaling pathway [54,55,56].

## 5. Conclusions

This study reveals the cytotoxic effect of 7-Epitaxol, the main active metabolite of paclitaxel, on HNSCC cells. Our findings reveal that 7-Epitaxol significantly reduces the viability of HNSCC cells by causing cell cycle arrest, as well as by inducing apoptotic and autophagic pathways. Regarding the molecular mechanism of 7-Epitaxol-induced cytotoxic effects, our findings reveal that the compound activates cell death pathways by suppressing the phosphorylation of ERK1/2 in HNSCC cells. In conclusion, this study identifies 7-Epitaxol as a potential chemotherapeutic agent that is known to have a higher stability and cytotoxicity than paclitaxel.

## Figures and Tables

**Figure 1 cells-10-02633-f001:**
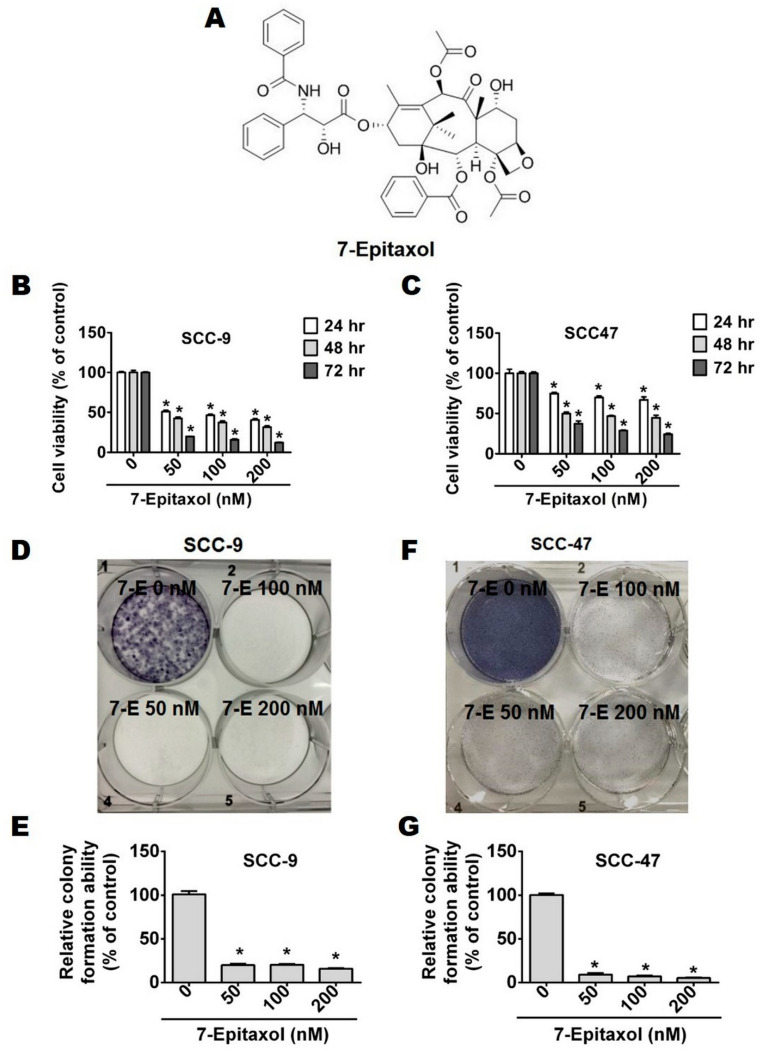
The cytotoxicity effects of 7-Epitaxol in SCC-9 and SCC-47 cell lines. (**A**) The chemical structure of 7-E. Cell viability was measured by MTT assay. (**B**) SCC-9 and (**C**) SCC-47 cells were treated with the indicated concentration of 7-E (0, 50, 100 and 200 nM) for 24, 48 and 72 h. (**D**,**E**) SCC-9 and (**F**,**G**) SCC-47 were analyzed by colony formation assay and cells were cultured in the condition medium presence of 7-E (0–200 nM) for 14 days. Data are presented as mean ± SD (*n* = 3). * *p* < 0.05, compared with the control group.

**Figure 2 cells-10-02633-f002:**
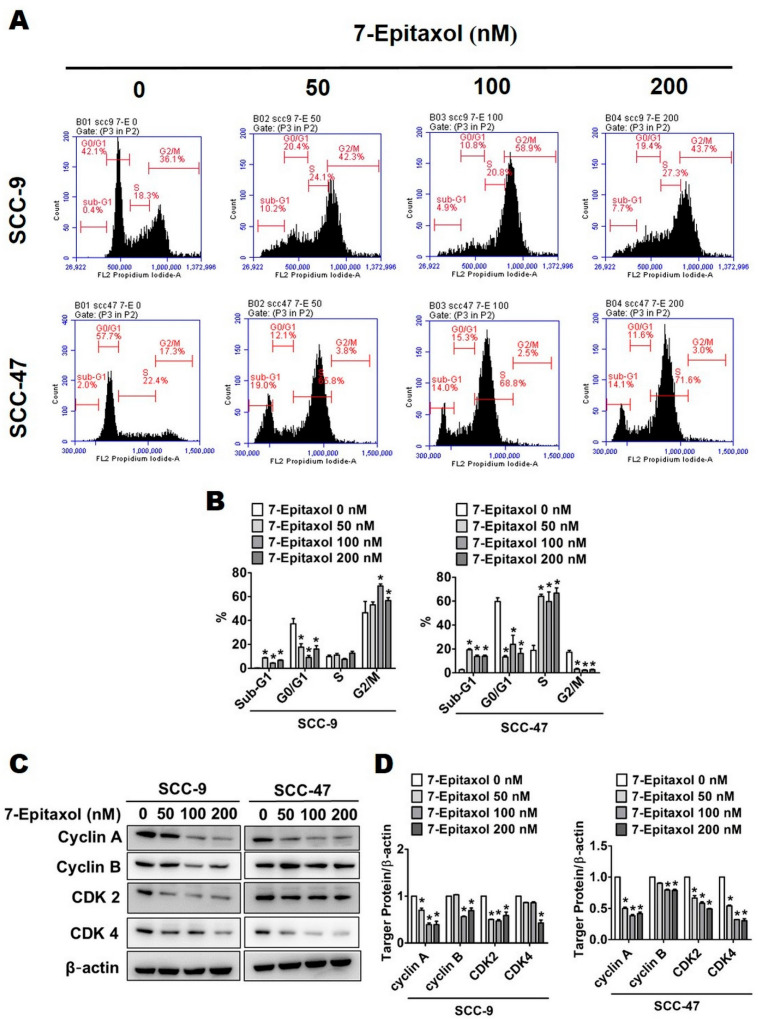
7-Epitaxol induces cell cycle arrest and apoptosis in SCC9 and SCC47 cells. After treatment with 7-E (0–200 nM) for 24 h: (**A**) Cells were PI stained and flow cytometry was performed to estimate cell cycle phase distribution. (**B**) Quantification of different cell cycle phase of sub-G1, G0/G1, S and G2/M. (**C**) We analyzed the expression of cell cycle control proteins, including cyclin A, cycle B, CDK 2, CDK 4, and β-actin by Western blot. (**D**) Quantitative relative density of each protein level was normalized to β-actin. Data are presented as mean ± SD (*n* = 3). * *p* < 0.05, compared with the control group.

**Figure 3 cells-10-02633-f003:**
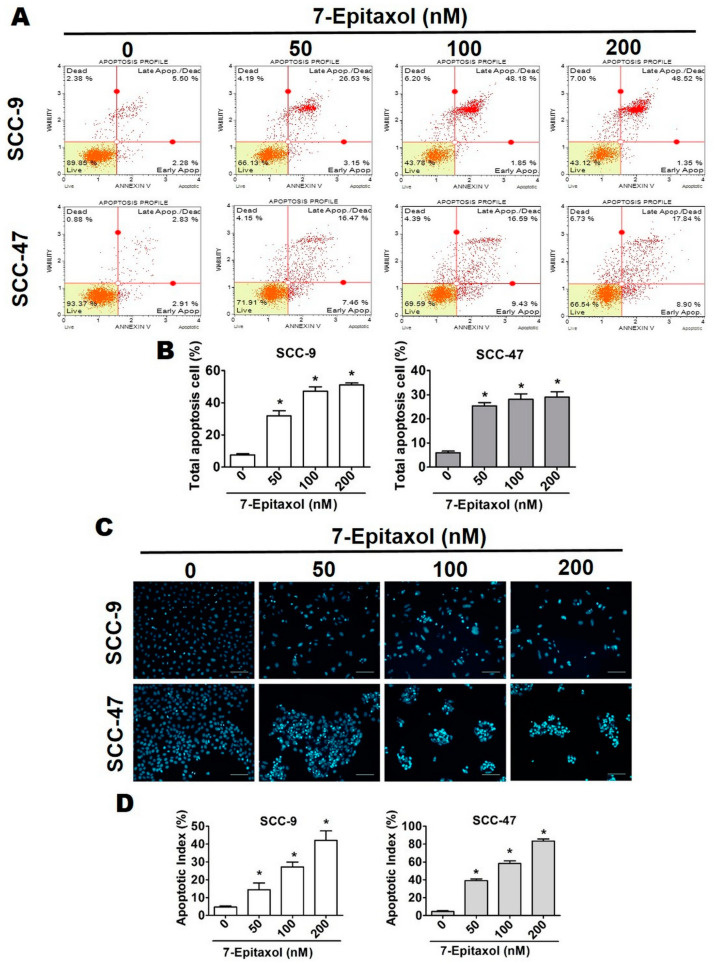
7-Epitaxol induces apoptosis in SCC-9 and SCC-49 cells. After treatment with 7-E (0–200 nM) for 24 h: (**A**) Cells were stained with Annexin V/PI and flow cytometry revealed 7-E induced apoptosis. (**B**) Quantitative relative percentages of apoptosis cells (including early and late states). (**C**,**D**) We used DAPI stain assay to determine DNA condensation with fluorescence microscopy. Bar scale = 100 µm. Data are presented as mean ± SD (*n* = 3). * *p* < 0.05, compared with the control group.

**Figure 4 cells-10-02633-f004:**
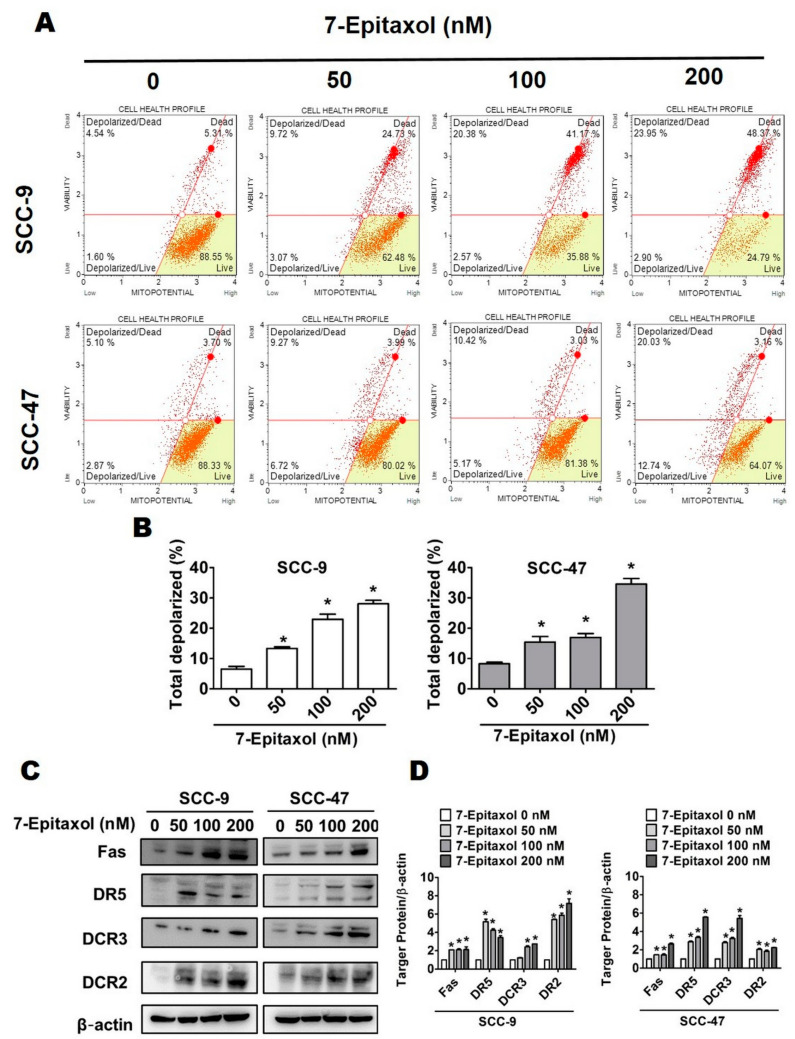
Intrinsic pathway and the extrinsic pathway were regulated by 7-Epitaxol in HNSCC cell lines. After treatment with 7-E (0-200 nM) for 24 h: (**A**) Mitochondrial membrane potential measurement assay was used with flow cytometry. (**B**) Data were analyzed by Muse Cell Analyzer (Millipore). (**C**) We analyzed the expression of intrinsic pathway control proteins, including Fas, DR5, DcR3, DcR2, and β-actin by Western blot. (**D**) Quantitative relative density of each protein level was normalized to β-actin. Data are presented as mean ± SD (*n* = 3). * *p* < 0.05, compared with the control group.

**Figure 5 cells-10-02633-f005:**
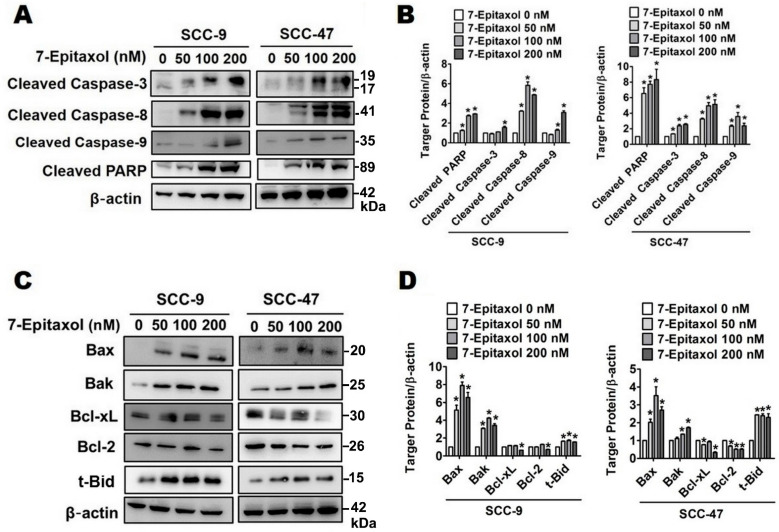
7-Epitaxol activates caspase pathway and regulates Bcl-2 family in SCC-9 and SCC-47 cells. Western blotting was used to measure the expression of regulated proteins after 24 h of 7-E treatment in (**A**,**B**) the caspase pathway related proteins and (**C**,**D**) the Bcl-2 family related proteins. Quantitative relative density of each protein level was normalized to β-actin. Data are presented as mean ± SD (*n* = 3). * *p* < 0.05, compared with the control group.

**Figure 6 cells-10-02633-f006:**
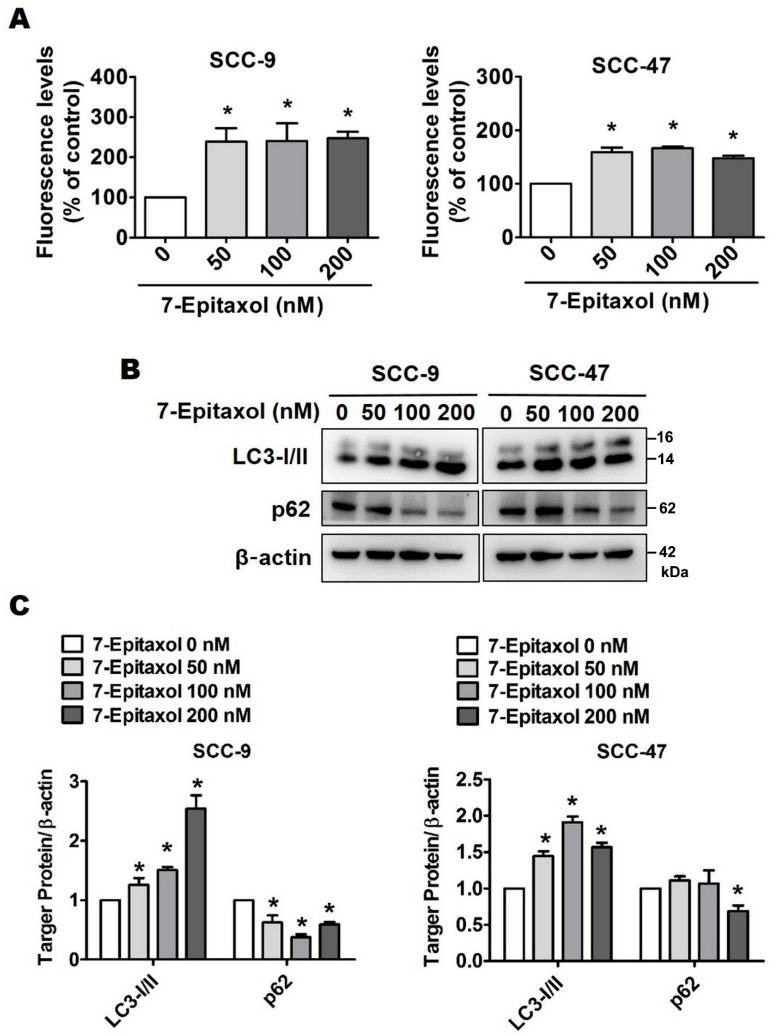
7-Epitaxol induces autophagy in SCC-9 and SCC-47 cells. After treatment with 7-E (0–200 nM) for 24 h: (**A**) Cells were used in a Cell Meter Autophagy Assay Kit to analyze the autophagy percentage with a fluorescence microplate reader. (**B**,**C**) Western blotting was used to measure the expression of regulated proteins including LC3-I/II and p62. Quantitative relative density of each protein level was normalized to β-actin. Data are presented as mean ± SD (*n* = 3). * *p* < 0.05, compared with the control group.

**Figure 7 cells-10-02633-f007:**
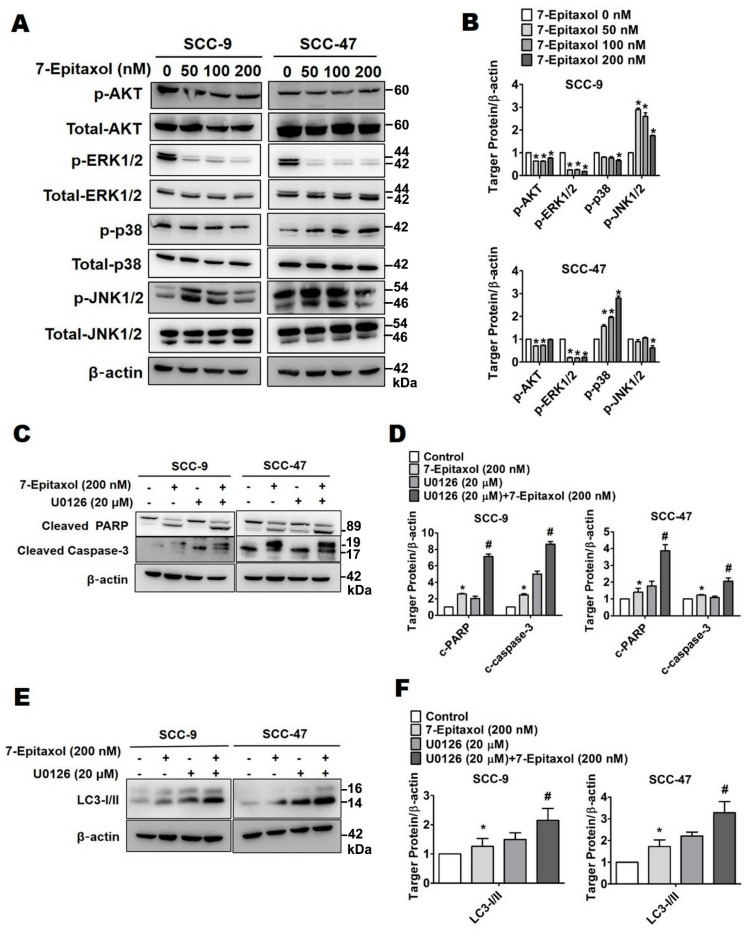
Epitaxol induces apoptosis and autophagy by affecting the AKT and MAPK pathways in HNSCC cell lines. Cell lines were pre-treated with or without U0126 for 1 h, then treated with 7-E for 24 h. Western blotting was used to measure the expression of regulated proteins (**A**,**B**) in the AKT and MAPK pathways (**C**,**D**) the caspase pathway, and (**E**,**F**) the related autophagy proteins. Quantitative relative density of each protein level was normalized to β-actin. Data are presented as mean ± SD (*n* = 3). * *p* < 0.05, compared with the control group. # *p* < 0.05, compared with the cells treated with 7-E (200 nM).

## Data Availability

Not applicable.

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
