# Peer review of "7-Epitaxol Induces Apoptosis and Autophagy in Head and Neck Squamous Cell Carcinoma through Inhibition of the ERK Pathway"

_cells, 2021, doi:10.3390/cells10102633_

Round 1

Reviewer 1 Report

The research performed by Kumar e colaboradores describes the anticancer effects of 7-Epitaxol towards Squamous cell carcinoma of the head and neck (HNSCC).

The manuscript is well written and organized. The authors also provided more insights about the mechanisms involved in 7-Epitaxol action: induction of apoptosis and autophagy through inhibition of ERK pathway.   For this reviewer, the paper deserves be published in this journal after minor adjustments. The suggestions are exposed in the attached pdf file.

Author Response

Answers: Thank you. We have improved the manuscript with your suggestions.

Reviewer 2 Report

The article presented by V. Bharath Kumar and collaborates, entitled “7-Epitaxol induces head and neck cell lines apoptosis and autophagy effects through inhibition of ERK pathway”, is a research paper that analyze the anticancer effects of 7-Epitaxol in two  carcinoma cell lines, SSC9 and SCC-47.  It is a basic research, which uses different biochemistry approaches.

The article is well written, easy to understand and the vocabulary is correct. The topic is within the scope of Cells.

The approach of the work is correct and the techniques used are adequate. The error I have detected is the absence of the control used. In material and methods, the authors comment that the drug is dissolved in DMSO, so the vehicle is supposed to be 0.2% DMSO - the authors do not refer to the % of DMSO used - but they should have used, in addition and in parallel, a control with water only in order to study the effect of DMSO.

The authors could also have included the drug paclitaxel as a positive control in order to compare both drugs. The authors should make some reference to the reason for the use of these concentrations (50, 100 and 200nM), comparing with those commonly used with paclitaxel.

The authors self-cite 0 times.

Minor revision.

-microliter with µ no with u

-line 185. The findings of MTT 185 assay revealed that the treatment with 7-E significantly reduced the cell viability in a dose- 186 and time-dependent manners, as compared to that in untreated control cells. It’s not a dose-dependent-

- Error bars figures 1E and 1G

- kDa in WB

- Scale bar Figure 3C

Author Response

Comments and Suggestions for Authors

The article presented by V. Bharath Kumar and collaborates, entitled “7-Epitaxol induces head and neck cell lines apoptosis and autophagy effects through inhibition of ERK pathway”, is a research paper that analyze the anticancer effects of 7-Epitaxol in two carcinoma cell lines, SSC9 and SCC-47. It is a basic research, which uses different biochemistry approaches.

The article is well written, easy to understand and the vocabulary is correct. The topic is within the scope of Cells.

The approach of the work is correct and the techniques used are adequate. The error I have detected is the absence of the control used. In material and methods, the authors comment that the drug is dissolved in DMSO, so the vehicle is supposed to be 0.2% DMSO - the authors do not refer to the % of DMSO used - but they should have used, in addition and in parallel, a control with water only in order to study the effect of DMSO.

Answers: Thank you for the suggestion. We have referred the percentage of DMSO used in materials and methods section. The vehicle group is present as 0 nM which contained less than 0.2% DMSO.

The authors could also have included the drug paclitaxel as a positive control in order to compare both drugs. The authors should make some reference to the reason for the use of these concentrations (50, 100 and 200nM), comparing with those commonly used with paclitaxel.

Answers: Thank you for the suggestions.
1. We have noticed that paclitaxel has been treated on HNSCC cells in previous study. (J Cancer Res Clin Oncol. 2016 Jun;142(6):1261-71) The study also mentioned that paclitaxel (10 µM) induced HNSCC cells apoptosis through the activation of caspase 3, 8 and 9, which gave evidences to compare with our study.
2. We have added the description in the result section (lines 186 to 188) to include this information: “The working concentrations of 7-E was based on previous study that treated paclitaxel on squamous carcinoma cells (
Oncol Lett. 2019 Sep;18(3):3195-3201.)”

The authors self-cite 0 times.

Minor revision.

-microliter with µ no with u

Answers: Thank you for the observation. We have improved errors in materials and methods section.

-line 185. The findings of MTT assay revealed that the treatment with 7-E significantly reduced the cell viability in a dose- and time-dependent manners, as compared to that in untreated control cells. It’s not a dose-dependent-

Answers: Thank you for the observation. We have removed the description in result section.

- Error bars figures 1E and 1G

Answers: Thank you for the observation. We have improved in figure 1E and 1G.

- kDa in WB

Answers: Thank you for the observation. We have referred the units of protein size in figure 4-7.

- Scale bar Figure 3C

Answers: Thank you for the observation. We have added scale bars in each figure of figure 3C (Bar scale=100 µm).

Round 2

Reviewer 2 Report

I believe the manuscript has been sufficiently improved to warrant publication in Cells.

Author Response

Thank you for the suggestions.
The abstract sentence has been modified (red words) in revise manuscript. 
Please see the attachment.
